# pH-Triggered Release Performance of Microcapsule-Based Inhibitor and Its Inhibition Effect on the Reinforcement Embedded in Mortar

**DOI:** 10.3390/ma14195517

**Published:** 2021-09-23

**Authors:** Jinzhen Huang, Yangyang Zhu, Yuwei Ma, Jie Hu, Haoliang Huang, Jiangxiong Wei, Qijun Yu

**Affiliations:** 1School of Materials Science and Engineering, South China University of Technology, Guangzhou 510640, China; msjinzhenh@mail.scut.edu.cn (J.H.); zhuyy10@vanke.com (Y.Z.); concyuq@scut.edu.cn (Q.Y.); 2Southern Marine Science and Engineering Guangdong Laboratory (Zhanjiang), Zhanjiang 524006, China; 3China Vanke Co., Ltd., Shenzhen 518020, China; 4Research Center for Wind Engineering and Engineering Vibration, Guangzhou University, Guangzhou 510006, China; yuwei_ma@gzhu.edu.cn

**Keywords:** microcapsule-based inhibitors, composition alterations, release behavior, mortar, pH sensitive, corrosion resistance

## Abstract

The smart release of healing agents is a key factor determining the inhibition efficiency of microcapsules-based corrosion inhibitors for reinforced concrete. In this study, the release behavior of benzotriazole (BTA) in microcapsule-based inhibitors was investigated in mortar sample to clarify the influence of different hydration products on the release process. The results indicated that under high pH environment (pH > 12.4), only about 5% reserved BTA was released from the mortar sample. pH drop resulted in the increased release of BTA from mortar sample. Most BTA in the microcapsule-based inhibitors was released from mortar sample in low pH environment, which was closely related to morphology/composition alterations of hydration products caused by pH drop of the environment. The smart release of BTA dramatically delayed corrosion initiation of reinforced mortar and halted corrosion product accumulation on the steel surface. Therefore, the corrosion resistance of the reinforced mortar was improved after corrosion initiation.

## 1. Introduction

Corrosion of the steel bar is widely accepted as the important factor causing the damage of reinforced concrete. Chlorides and carbonation are two main reasons initiating corrosion of the steel bar. Carbonation causes the neutralization of cement-based materials and subsequently pH drop from 13 to about 8–9 [1]; pit corrosion can be initiated when sufficient chlorides penetrate at the reinforcing steel/concrete interface, and pH at corrosion sites is lower than 5 [1,2], leading to more serious corrosion damage and reduced service life of reinforcement concrete [3,4].

The addition of organic corrosion inhibitors is an efficient way to protect reinforced concrete from corrosion damage [5,6,7]. Normally, organic corrosion inhibitors form an adsorption film on the reinforcing steel surface, reducing anode or cathode reaction rate during corrosion process and subsequently improving the corrosion resistance of the reinforcing steel [8]. However, some adverse impacts of organic corrosion inhibitors were reported when used in concrete. For example, corrosion inhibitors based on amine and ester increased the pore size and subsequently water permeability of concrete [9], which would reduce the durability of reinforced concrete. Further, because the traditional organic corrosion inhibitors are not sensitive to environment alterations (i.e., pH reduction or Cl^−^ content increase) at the local corrosion sites, the corrosion damage of the reinforcement is not able to be targeted repaired immediately after corrosion initiation.

In recent years, microcapsules carrying with self-healing agents were proposed for achieving smart corrosion protection of reinforced concrete [10,11,12,13,14]. For example, microcapsules with calcium hydroxide core and ethocel shell maintained high OH^−^ concentration in cement pore solution due to the accelerated release (about 10% higher) of the encapsulated Ca(OH)_2_ when pH decreased, thus halting corrosion of the immersed reinforcing steel [11,14]. It was also reported that the released amount of sodium nitrite or sodium monofluorophosphate from microcapsules with ethocel or polystyrene resin shell was increased by 10% in low pH simulated cement pore solution (SPS) [10,15]. Liu et al. [16] prepared CaCO_3_ microparticles loaded with sodium lignosulfonate (SLS). The results indicated that the loaded SLS was released under different pH conditions of the aqueous media; however, SLS release amount in acidic testing solution (pH = 4) was about 50−80% higher than in alkaline testing solution (pH = 10). Ress et al. [17] synthesized microcapsules with NaNO_2_ core and colophony shell for corrosion protection of reinforced concrete to avoid the leaching of NaNO_2_ from concrete admixture. Pronounced pH sensitivity was observed for the prepared microcapsules: the released amount of NaNO_2_ was doubled in alkaline simulated concrete pore solution, compared to neutral Di water. Even though the above microcapsules avoid the leaching of NaNO_2_, its application for smart corrosion control of reinforced concrete should be further clarified, because the corrosion damage of the reinforcement is accompanied by pH drop from 13 to about 8–9. Furthermore, chloride-triggered microcapsules with Ca(OH)_2_ core and cross-linked poly-ionic liquids shell were also reported [18]. The presence of chloride in the testing solution resulted in a large amount of nanometer-sized holes on the shell, accelerating the release of Ca(OH)_2_. It can be found that the above proposed microcapsules exhibit pH sensitivity, thus are good candidates as the self-healing materials for reinforced concrete. However, pH sensitivity of the microcapsules can still be potentially increased.

In our previous studies [19,20], tailored microcapsule-based corrosion inhibitors (MCI) with no significantly harmful impact on cement-based materials was successfully prepared. Very high pH sensitivity was relevant for the prepared MCI: on one hand, the prepared MCI presented a high stability in SPS with pH of 13; on the other hand, 5 times higher amount of the released benzotriazole (BTA) was observed in SPS with pH lower than 11. Cement-based materials are much more complicated than SPS, and different hydration products will also significantly affect the release behavior of healing agents encapsulated in MCI. Therefore, it is of great importance to verify its pH sensitivity in cement-based materials before the above microcapsule-based corrosion inhibitors are used for smart corrosion control of reinforced concrete. To this end, the release performance of healing agents encapsulated in MCI was extensively characterized in mortar, and the relationship between the release behavior and composition/microstructure alterations of hydration products was clarified in this present study. Further, the corrosion inhibition effect of MCI on the reinforcement embedded in mortar was also evaluated.

## 2. Materials and Methods

### 2.1. Materials

Amphiphilic polyethylene oxide (PEO_113_)-b-polystyrene (PS_1171_) copolymer was used to prepare microcapsule-based corrosion inhibitor (MCI) by dialysis in this study; the details related to the preparation procedure, copolymers and MCI can be found in [19,20]. The concentration of benzotriazole (BTA) encapsulated in MCI was 2.10 mg/mg and the concentration of MCI solution was 0.5 g/L the average diameter of MCI was about 250 nm [19,20].

Simulated cement pore solution (SPS) in this study contained 0.06 mol/l sodium hydroxide and saturated calcium hydroxide (with pH of 12.7) [7]. In order to clarify the influence of pH on the release amount of BTA reserved in MCI, pH of SPS was changed to 12.4, 11.5, 9 and 7, respectively.

Both mortar sample and reinforced mortar samples with dimensions of 40 mm×40 mm × 160 mm were cast by using PII 42.5 ordinary Portland cement with the chemical and mineral compositions as shown in Table 1 and Table 2, respectively. D_50_ (maximum particle size when accumulative volume of cement reached 50 vol. %) of the used cement was 24 µm in this study (as shown in Figure 1). For reinforced mortar, the used reinforcement was HPB235 construction reinforcement (the chemical compositions are presented in Table 3); the diameter and working area of the used reinforcement was 8 mm and 35.2 cm^2^, respectively. The water to cement (*w/c*) ratio and cement to sand (*c/s*) ratio for (reinforced) mortar were 0.5 and 1:3. For mortar samples, the mixing water was MCI solution. For reinforced mortar samples, the mixing water was deionized water for the reference MCI-free sample and MCI solution for the MCI-containing sample. As a result, MCI concentration in (reinforced) mortar was 0.025 wt. % (per dry cement weight); correspondingly, BTA concentration in (reinforced) mortar was 0.0525 wt. % (per dry cement weight). The curing condition for (reinforced) mortar sample was 20 °C and 95% RH. At 28 days, mortar samples were crushed into small pieces and then ground into fine powders. 1.0 g mortar powders were immersed in 20 mL SPS with difference pH values for 3 d; centrifugation was then conducted with the relevant SPS for 10 min at 4800 rpm. The collected solution by filtering with 0.45 µm filter film was used to characterize the released BTA amount in SPS. Further, the mortar powders collected after centrifugation were used to investigate the alterations on morphology and compositions of hydration products in mortar samples. Before the above tests, the collected mortar powders were immersed in ethanol for 48 h to stop hydration. In our previous study [19], after PEO-b-PS copolymers were synthesized, the products were purified by precipitating in methanol for three times before the preparation of MCI. As a result, it was believed that PEO-b-PS copolymers and MCI should be very stable in ethanol and the immersion in ethanol should not exhibit obvious effect on MCI in this present study. After immersion, the mortar powders were separated from ethanol by filtration with 0.45 µm filter film and dried at 40 °C in vacuum drying chamber until constant weight. After curing for 28 days, reinforced mortar samples were half soaked in 3.5 wt. % NaCl solution for further investigations.

### 2.2. Methods

#### 2.2.1. Compressive Strength and Flexural Strength

The dimensions of mortar samples for compressive strength and flexural strength tests were 40 × 40 × 40 mm^3^ and 40 × 40 × 160 mm^3^, respectively. The measurements of mechanical properties were conducted at the hydration age of 1 d, 3 d, 7 d and 28 d, respectively. There were 3 replicates for each specimen.

#### 2.2.2. Release Amount of BTA Encapsulated in MCI

In this study, UV-vis spectrophotometry (HEλIOS Gamma and Delta, Thermo Scientific, Waltham, MA, USA) was used to evaluate the released amount of BTA in SPS with a wavelength of 265 nm. The background value for UV-vis spectrophotometry tests was determined by measuring the absorbance of SPS without MCI. The released percentage of BTA was expressed in Equation (1):
(1)$$M_{r}\;=\;C_{0}V_{s}/M_{m}$$
where *M_r_* is the released percentage of BTA; *C*_0_ is the measured BTA concentration in SPS; vs. is SPS volume used for the immersion of mortar sample; *M_m_* is BTA content in 1.0 g mortar sample.

#### 2.2.3. Morphology and Composition of the Mortar Samples

In this study, scanning electronic microscope (SEM, EVO18, Zeiss, Oberkochen, Germany) was used for morphology observations of mortar sample after 3 days immersion with the magnifications of 100× and 5000× (10 kV in second electronic (SE) mode). Both EDS (Merlin Compact VP, Oxford, Abingdon, England) and XRF (Axios Pw 4400, PANalytical B.V., Almelo, Netherland) were applied to evaluate the chemical compositions of mortar sample. The capacity of X-ray tube at Rhodium (Rh) target window was 4 kW and the accuracy was 0.05% for XRF analysis.

The mineral composition of mortar sample after immersed in SPS for 3 days was determined by XRD analysis (X’pert PRO, PANalytical B.V., Almelo, The Netherland). XRD analysis with energy source of Cu K_a_ (40 kV and 40 mA) was conducted between 5–90° with scan rate of 10°/min and wavelength of 0.15 nm. The main function groups (i.e., Al-O and Si-O) of hydration products in mortar samples were examined by FTIR (Nicolet Nexus for Euro, Thermo Scientific, Waltham, MA, USA) in this study. The scan number was 32 with the collected wavenumber between 400 cm^−1^ and 4000 cm^−1^. The resolution of FTIR tests was 0.4 cm^−1^. The polymerization degree of ^29^Si and ^27^Al for hydration products in mortar samples was investigated by NMR (AVANCE III HD 600, Bruker, Karlsruhe, Germany). The detector used for NMR tests was CP/MAS solid detector (Bruker, Karlsruhe, Germany). The used magnetic intensity was 400 T and the magic angle rotation rate was 15 kHz. The scanning number for NMR tests was 128.

#### 2.2.4. Electrochemical Behavior of Reinforced Mortar

The electrochemical behavior of reinforced mortar was characterized by open circuit potentials (OCP) and potentio-dynamic polarization (PD) at different immersion time intervals (21 d, 72 d, 85 d, 92 d, 115 d and 135 d). As three-electrode set-up, the reference electrode, working electrode and counter electrode was saturated calomel electrode (SCE), reinforcement electrode and Ti mesh, respectively. At 25 ± 1 °C, PD (Metrohm Autolab-Potentiostat PGSTAT 302N) was tested from −200 mV to +1000 mV vs. OCP of the reinforcing steel with the scan rate of 0.5 mV/s.

#### 2.2.5. Surface Analysis of the Reinforcement Embedded in Mortar

After 135 d, reinforced mortar samples were vacuum dried at 60 °C for 1 day. The reinforced mortar was then broken and the morphology of the reinforcement surface was examined by SEM observations. 20 kV was used as the accelerating voltage for SEM observations and images (in SE mode, with a magnification of 500×) of the reinforcement surface were derived. Further, backscattered electronic (BSE) images (magnification of 200×) at the reinforcement/mortar interface were also obtained to characterize the accumulation of corrosion products after the treatment.

The chemical composition of the corrosion products was characterized by Raman analysis (LabRAM Aramis, HORIBA JobinYvon S.A.S, Kyoto, Japan) with magnification of 50× and power of 0.5 mW. The wavelength of the used laser and cumulative scanning number for Raman analysis was 514.5 nm and 200, respectively. The exposure time for Raman analysis was 1 s.

## 3. Results

### 3.1. Mechanical Properties of Mortar Sample in the Presence of MCI

Figure 2 shows the compressive strength and flexural strength of mortar samples at different hydration ages. It was observed that MCI-containing mortar sample exhibited similar compressive strength at early hydration ages (1 d, 3 d and 7 d). At 28 days, the compressive strength of the mortar sample was slightly increased in the presence of MCI: the compressive strength of MCI-free and MCI-containing mortar sample was about 60 MPa and 70 MPa, respectively. MCI exhibited no obvious influence on the flexural strength of mortar sample in this present study. It indicated that the addition of MCI presented no negative effect on the mechanical properties of mortar samples.

### 3.2. Release amount of BTA Encapsulated in MCI from Mortar Sample

Figure 3 presents the release percentage of the encapsulated BTA from the mortar sample after 3 days immersion in different SPS. Generally, the release percentage of BTA increased when pH of SPS was reduced. After immersed in SPS with pH of 12.7 (reference sample) and 12.4, the release percentage of the encapsulated BTA was very low, and only about 5% BTA was released from the mortar sample. A slightly higher release percentage of BTA (10%) was observed when pH of SPS decreased to 11.5. In SPS with pH of 9, the release percentage of BTA was increased to about 25%. When pH of SPS was further reduced to 7, BTA presented a very high release percentage of 95%. Therefore, pH-triggered release performance of MCI was confirmed by the above experimental results. When pH was higher than 11.5, the released BTA content was very low, indicating that BTA steadily existed in mortar sample under high alkaline environment. When pH was lower than 9, BTA was released with a significantly higher rate; specifically, when pH was reduced to 7, almost all BTA reserved in MCI was released from the mortar sample.

### 3.3. Morphology Alterations of the Mortar Samples

Figure 4 shows SEM images of mortar samples immersed in different SPS for 3 days. For the reference sample (before immersion, Figure 4a), both sands with large dimensions (red circles in the image) and hydration products with small dimensions (blue circles in the image) were observed. In SPS with pH of 12.4 (Figure 4b), the above mentioned sands and hydration products also existed in mortar sample; however, the amount of hydration products was slightly lower, compared to the reference sample. Further, more needle shape hydration products were observed in the sample immersed in SPS with pH of 12.4. When pH of SPS was reduced to 7 (Figure 4c), the amount of hydration products was further reduced and mainly sands with large dimensions were observed in SEM image. This might be related to the dissolution of hydration products in SPS with low pH value, which will be further confirmed by XRF, XRD, FTIR and NMR analysis in Section 3.4. As a result, the hydration products exhibited more porous microstructure, compared to both reference sample and sample immersed in SPS with pH of 12.4, evidenced by the enlarged SEM images in Figure 4. The above morphology alterations of the mortar samples immersed in different SPS was attributed to the influence of pH on the composition of hydration products in mortar matrix, which is discussed further below.

### 3.4. Composition Alterations of the Hydration Products in Mortar Samples after Immersed in SPS

XRF results of the mortar sample after immersed in SPS with different pH for 3 d are presented in Table 4. For the reference sample, the main chemical composition of the mortar sample included SiO_2_, CaO, Al_2_O_3_ and Fe_2_O_3_. SiO_2_ content was higher than 80 wt. %, which was mainly corresponding to the sands in mortar sample. The chemical composition of the mortar sample after immersed in SPS with pH of 12.7 was very similar to the reference sample. After immersed in SPS with lower pH (i.e., pH of 12.4, 11.5, 9 and 7), the chemical composition of the mortar sample was altered: SiO_2_ content increased and CaO content decreased. Further, a lower pH of SPS resulted in a more significant alteration on the chemical composition of the mortar sample. For example, SiO_2_ and CaO content for the reference sample was 82.14 wt. % and 12.26 wt. %, respectively. However, after immersed in SPS with pH of 11.5, SiO_2_ content increased to 89.02 wt. % and CaO content was reduced to 6.04 wt. %; after immersed in SPS with pH of 7, SiO_2_ content further increased to 91.97 wt. % and CaO content was only 2.93 wt. %. XRF results indicated that the immersion in SPS solution with low pH caused decomposition and dissolution of hydration products, leading to the increase of SiO_2_ content and reduction of CaO content in mortar sample.

Figure 5 presents XRD results of the mortar samples in different SPS at 3 days. Before immersion (the reference sample), the main crystal phases in the mortar sample were quartz (SiO_2_), calcium hydroxide (CH) and hemicarboaluminate hydrate (Hc). When immersed in SPS with pH of 12.7, the crystal phases were quite similar to the reference sample. When immersed in SPS with pH of 12.4, the intensity of peaks corresponding to CH was significantly reduced; the crystal phases were mainly consisted of SiO_2_ and Hc. When immersed in SPS with pH of 11.5, besides CH, the peaks related to Hc also disappeared; the main crystal phase was only SiO_2_ corresponding to sands in the mortar sample. When the pH of SPS decreased to 9, a new peak at 2θ 27º appeared which was corresponding to CASH [22]; when pH of SPS was further reduced to 7, the peak intensity of CASH was increased. The above results indicated that the decrease on pH of SPS resulted in the dissolution of CH and Hc and generation of CASH gel.

Figure 6 shows FTIR results of mortar samples after immersed in SPS with different pH for 3 days. Before immersion (reference sample), monosilicates (Q^0^, at 970 cm^−1^), disilicates and chain end groups (Q^1^, at 812 cm^−1^) and middle groups in chains (Q^2^, at 967–1060 cm^−1^) [23,24,25,26] existed in the mortar sample; further, the infrared peaks of C=O bond for CO_3_^2−^ (875 and 1400–1500 cm^−1^ [25,27]) were also observed for the reference sample. FTIR results for the mortar samples immersed in SPS with pH of 12.7, 12.4 and 11.5 were quite similar to the reference samples before immersion, and the infrared peaks corresponding to Q^1^, Q^2^ and C=O bond also existed. However, for the mortar samples immersed in SPS with pH of 9 and 7, the infrared peak corresponding to Q^1^ was reduced and the infrared peak related to Q^2^ shifted to higher wavenumber, indicating the higher polymerization degree of silica chain for mortar samples in these solutions [28]. In addition, the infrared peaks for C=O bond was shifted to lower wavenumber for the mortar samples immersed in SPS with pH of 11.5, 9 and 7, indicating that carbonate was decomposed when the mortar sample was immersed in SPS with pH < 11.5. Therefore, the pH threshold for the composition alteration of CSH was about 9. When pH was higher than 9, CSH was quite stable and its composition was not significant altered. When pH was below 9, the polymerization degree of CSH increased, indicating that CSH was decomposed in mortar sample. Further, when pH was higher than 11.5, the carbonates in the mortar sample was very stable; when pH was lower than 11.5, carbonates became decomposed.

^29^Si and ^27^Al NMR spectra of the mortar sample after immersed in SPS with different pH for 3 d are presented in Figure 7 and Figure 8 to further confirm the composition alterations of hydration products. Table 5 summarizes the chemical shifts of Q^0^, Q^1^, Q^2^, Q^3^ and Q^4^ in ^29^Si NMR spectrum for cement-based materials [29]. Compared to the reference sample, when immersed in SPS with pH of 12.4, the intensity of the peaks corresponding to Q^0^ was reduced and the intensity of the peaks related to Q^2^ increased in ^29^Si NMR spectra (Figure 7). When immersed in SPS with pH of 7, the peaks corresponding to Q^0^ and Q^1^ almost disappeared; the peak corresponding to Q^2^ was significantly shifted to the right direction and the peaks related to Q^4^ were not obviously altered. ^29^Si NMR results confirmed that CSH was decomposed in SPS with low pH value. Based on ^27^Al NMR spectra (Figure 8), the intensity of the peaks corresponding to Al-O bond was not obviously altered for the sample immersed in SPS with pH of 12.4, compared to the reference sample. In SPS with pH of 7, the peaks corresponding to Al-O bond in AFt and AFm (about 13 ppm and 8ppm, respectively [29]) almost disappeared, indicating the decomposition of AFt and AFm; the intensity of the peak related to Al-O bond connecting with silica increased. Further, a new Al-O bond appeared at about 5 ppm which might be related to hydrous alumina gel [30], hydrated calcium aluminate [31,32] or aluminum hydroxide with low crystallinity [33].

### 3.5. Corrosion Inhibition Effect of MCI on the Reinforcement Embedded in Mortar

#### 3.5.1. Influence of MCI on the Electrochemical Behavior of the Reinforcement Embedded in Mortar

Open circuit potential (OCP) of the reinforcement embedded in mortar samples is presented in Figure 9. OCP of both the reference and MCI-containing samples exhibited very positive value (in the range of −110 mV to −212 mV) at early immersion age; OCP of MCI-containing sample was slightly more negative than the reference sample. OCP of all samples were more positive than −250 mV (OCP threshold for passivity of the reinforcement [34]), indicating that the reinforcement embedded in mortar was in passive state both for the reference and MCI-containing samples at early immersion age. For the reference sample, OCP of the reinforcement was shifted to more negative than −400 mV after immersed for 80 d, indicating corrosion initiation of the reinforcement at this moment. However, MCI-containing sample still presented OCP of about −160 mV at the same time interval. For MCI-containing sample, the time for negative OCP shift of the reinforcement was prolonged to about 113 days and its potential was negatively shifted to about −500 mV while OCP of the reference sample was about −460 mV. Therefore, corrosion initiation of the reinforcing steel embedded in mortar was significantly retarded in the presence of MCI. After corrosion initiation, OCP of MCI-containing sample was more negative than the reference sample (about 40–120 mV difference); further, positive OCP shift was observed for MCI-containing sample at the immersion age of about 126 d. The above positive OCP shift for MCI-containing sample might be related to self-repairing of the reinforcement surface due to the release of BTA from MCI at the corrosion locations, which will be discussed in detail further below.

PD curves of the reinforced mortar samples are shown in Figure 10. At early immersion ages (i.e., 21 days and 72 days, as shown in Figure 10a), PD curves for the reference and MCI-containing samples were almost identical; the reference sample exhibited slightly more positive corrosion potential (E_corr_) and lower anodic current density, compared to MCI-containing sample. The above results indicated that MCI didn’t present pronounced effect on the electrochemical behavior of the embedded reinforcement before corrosion initiation. At 85 d and 92 d (Figure 10b), based on OCP results in Figure 9, corrosion damage for the reference sample was initiated; however, MCI-containing sample was still in passive state. As a result, E_corr_ for MCI-containing sample was dramatically more positive, compared to the reference sample; the anode current density for MCI-containing sample was also significantly lower than the reference sample. MCI exhibited efficient corrosion inhibition effect on the reinforcement embedded in mortar during this immersion stage. At late immersion ages of 115 d and 135 d (Figure 10c), both the reference and MCI-containing samples exhibited very negative E_corr_ value, which was corresponding to corrosion damage of the embedded reinforcement. At this stage, E_corr_ for MCI-containing sample was more negative, compared to the reference sample. However, the anode and cathode current densities for MCI-containing sample were lower than the reference sample, indicating that MCI was the mixed type corrosion inhibitor [35] and still exhibited corrosion inhibition effect on the embedded reinforcement after corrosion initiation.

Corrosion current densities (I_corr_) for different samples calculated based on PD curves [36] are presented in Figure 11. At early immersion ages (i.e., 21 d and 72 d), the reinforcement for all samples was in passive state, thus exhibiting very low I_corr_ (in the range of 0.042 to 0.061 μA·cm^−2^). MCI didn’t present obvious influence on I_corr_. At 85 d and 92 d, I_corr_ for the reference sample was dramatically increased (in the range of 0.484 to 0.497 μA·cm^−2^); I_corr_ for MCI-containing sample was still maintained at very low level (in the range of 0.0416 to 0.0443 μA·cm^−2^) because the reinforcement was still in passive state. At late immersion age (i.e., 115 d and 135 d), because corrosion damage for MCI-containing sample was also initiated, I_corr_ was slightly increased (in the range of 0.114 to 0.169 μA·cm^−2^). However, I_corr_ for MCI-containing sample was still significantly lower, compared to the reference sample.

#### 3.5.2. Influence of MCI on the Morphology and Composition of Corrosion Products

Figure 12 shows SEM images at the reinforcement/mortar interface after 135 days. Severe accumulation of corrosion products was observed for the reference MCI-free sample, evidenced by a thickness of about 120 μm (Figure 12a). In the presence of MCI, the thickness of the formed corrosion products was dramatically reduced to about 20 μm. Figure 13 presents SEM images of different samples after 135 days. Corrosion products with porous microstructure and large size were observed for MCI-free sample (Figure 13a). For MCI-containing sample (Figure 13b), much less corrosion products with smaller size and more compact microstructure was observed on the reinforcement surface.

Raman spectra of the formed corrosion products for different samples after immersed in NaCl solution for 135 days are presented in Figure 14. For the reference sample, the mainly formed corrosion products were the mixtures of α-Fe_2_O_3_ and γ-FeOOH [37]. The above corrosion products exhibited porous microstructure and large volume expansion, leading to more serious corrosion damage of the steel bar [38]. MCI significantly halted the formation of corrosion products, evidenced by the dramatically lower peak intensity in Raman spectrum for MCI-containing sample. The main corrosion product for MCI-containing sample was γ-Fe_2_O_3_ with compact structure [39]. The above surface analysis results indicated that MCI significantly retarded corrosion product propagation on the surface of the reinforcement embedded in mortar sample.

## 4. Discussion

It was mentioned in Introduction section that our previous study [19] proved the pH sensitivity for the release of BTA encapsulated in MCI in simulated cement pore solution. In SPS with pH of 13, BTA was steadily loaded in MCI; when pH of SPS decreased to 11, 5 times more BTA was released from MCI. The pronounced pH sensitivity for BTA release in SPS was possibly related to the hindered diffusion of BTA resulted from the generation of Na rich film around MCI and agglomerates of MCI under high alkaline environment (pH = 13). In this present study, MCI also exhibited high pH sensitivity in the mortar sample (Figure 3). However, due to the small size and hydrophilic characteristic, MCI was covered by hydration products in mortar sample. Therefore, the release behavior of BTA in mortar sample was different from SPS, which was closely related to the composition/microstructure alterations of the hydration products under different pH environment (Figure 15).

When mortar sample was immersed in SPS with pH of 12.7 (pH of saturated Ca(OH)_2_ solution [40]), most hydration products were quite stable under this high alkaline environment (Figure 5 and Figure 6). When immersed in SPS with pH of 12.4, CH dissolved because pH of SPS was below the pH of saturated calcium hydroxide solution [40]. The decomposition and dissolution of other hydration products were not relevant (Figure 5, Figure 6, Figure 7 and Figure 8). Under the above high alkaline environment, a very low release rate was relevant for BTA from MCI, due to the previously reported retarded diffusion of BTA under high alkaline environment [19]. Since crystal hydration products, i.e., CH and AFt normally exhibited plate-like or needle-like shape [41], it was supposed that MCI was mainly covered by CSH in mortar sample, thus the dissolution of CH didn’t obviously influence the diffusion of BTA in the mortar sample. When pH of SPS was higher than 12.4, CSH stably existed in mortar sample, evidenced by FTIR and NMR results (Figure 6 and Figure 7). The stable CSH gel surrounding MCI further halted the diffusion of BTA in mortar sample. As a result, due to the combined effects of both high alkaline environment and barrier of the surrounded stable CSH gel, the released BTA amount from mortar sample was quite low (only about 5%) in SPS with pH > 12.4 (Figure 3 and Figure 15a,b).

When immersed in SPS with pH of 11.5, the reserved BTA was rapidly released from MCI, because MCI agglomerates and Na-rich film were not relevant under this pH environment according to our previous study [19]. However, CSH gel still stably existed in the mortar sample (Figure 6) because this pH value was higher than the critical pH for CSH decomposition (pH = 9 [40]). Therefore, BTA release rate was also very slow when mortar sample was immersed in SPS with pH of 11.5. In addition to CH, Hc (resulted from the carbonation of AFt) also decomposed in SPS with pH of 11.5 (Figure 5), leading to more porous hydration products around MCI. As a result, slightly more BTA (10%) was released from mortar into SPS with pH of 11.5 (Figure 15c), compared to SPS with pH of 12.7 and 12.4. This was quite different from BTA release behavior in SPS, in which the released BTA amount from MCI was increased by 5 times in SPS with pH < 11 [19]. When the mortar sample was immersed in SPS with pH of 9, CH and Hc were almost dissolved and decomposed in the mortar sample (Figure 5). Furthermore, CSH gel was also partially decomposed, evidenced by FTIR results (Figure 6). Even though a small amount of CASH formed by the reaction between silicate gel and hydrated calcium aluminate, evidenced by FTIR and ^27^Al NMR spectra (Figure 6 and Figure 8), the partial decomposition of CSH still led to more porous microstructure and reduced barrier effect of CSH around MCI. As a result, the release amount of BTA from the mortar sample was increased to 25% when immersed in SPS with pH of 9 (Figure 15d). When pH of SPS was further reduced to 7, more CSH in the immersed mortar sample was decomposed, evidenced by SEM observation, FTIR and NMR results (Figure 4c, Figure 6 and Figure 7). Therefore, the barrier effect of CSH around MCI was very limited, and most BTA (more than 95%) was released from the mortar sample immersed in SPS with pH of 7 (Figure 15e).

In this study, hydration products indeed significantly affected the release behavior of BTA loaded in MCI, leading to a more complicated BTA release process in mortar sample. However, the rapid release of BTA can be also achieved in mortar sample by pH drop which would happen at corrosion sites on the reinforcing steel. In reinforced mortar, before corrosion initiation of the reinforcing steel, BTA was stably reserved in MCI due to the high alkalinity in mortar. As a result, the corrosion resistance was not obviously influenced by MCI (Figure 10a and Figure 11). However, the admixed MCI adsorbed on the surface of steel bar, hindering O_2_ and chloride diffusions and subsequently leading to slightly negative shift of OCP and E_corr_ at this stage and retarded corrosion initiation for MCI-containing sample (Figure 9 and Figure 10b). After corrosion initiation, BTA reserved in MCI was rapidly released because of pH drop at corrosion sites on the reinforcing steel surface. The released BTA together with the empty MCI acted as the barrier, halting both anode and cathode reactions during the corrosion damage process, thus inhibiting corrosion damage of the reinforcing steel embedded in mortar after corrosion initiation (Figure 10c and Figure 11). The inhibition effect of MCI was also evidenced by the pronounced positive OCP shift (about 400 mV) for MCI-containing sample after 126 d immersion (as shown in Figure 9). The corrosion inhibition effect of MCI on reinforced mortar reported in this present study confirms that the rapid release of BTA in the mortar sample caused by pH drop makes the prepared MCI possesses a high potential application on smart corrosion protection, increasing the corrosion resistance of reinforced concrete. The corrosion inhibition effect and related mechanisms of MCI on reinforced mortar was also extensively investigated and will be separately reported elsewhere.

## 5. Conclusions

The release behavior of the reserved BTA in MCI exhibited pronounced pH sensitivity in mortar sample in this paper. Under high pH environment (pH > 12.4), only about 5% reserved BTA was released from the mortar sample; when pH was reduced to 11.5 and 9, the release amount of BTA from the mortar sample was increased to about 10% and 25%, respectively. Under low pH environment (pH = 7), most BTA (more than 95%) was released from the mortar sample.

The composition alterations of the hydration products in the mortar sample were also observed when immersed in SPS with different pH value. When the pH of SPS decreased, CH and Hc were first decomposed; when the pH of SPS was lower than 9, CSH in mortar sample became decompose and a small amount of CASH formed. The above composition alterations of the mortar sample resulted in more porous hydration products surrounding MCI, leading to faster BTA release from the mortar sample under low pH environment.

Due to the smart release of BTA in mortar sample, MCI efficiently retarded corrosion initiation of reinforced mortar. Further, the released BTA significantly reduced corrosion product accumulation, thus dramatically increased corrosion resistance of the reinforcement after corrosion initiation.

## Figures and Tables

**Figure 1 materials-14-05517-f001:**
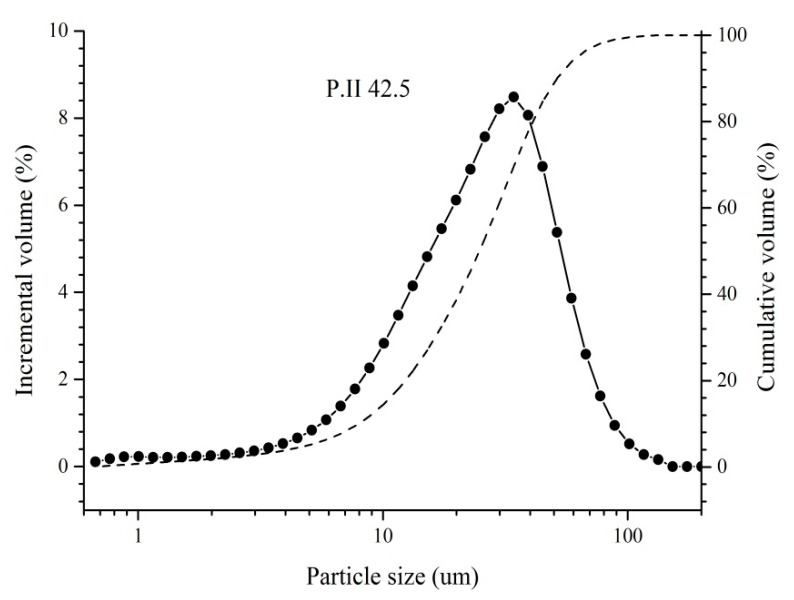
Particle size distribution of P·II 42.5 Portland cement used in this study.

**Figure 2 materials-14-05517-f002:**
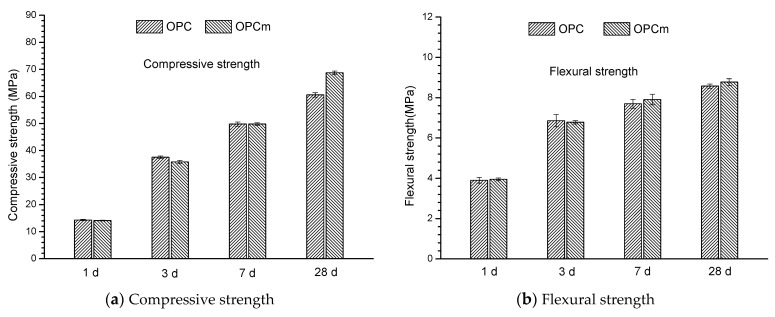
Compressive strength (**a**) and flexural strength (**b**) of mortar samples at different hydration ages.

**Figure 3 materials-14-05517-f003:**
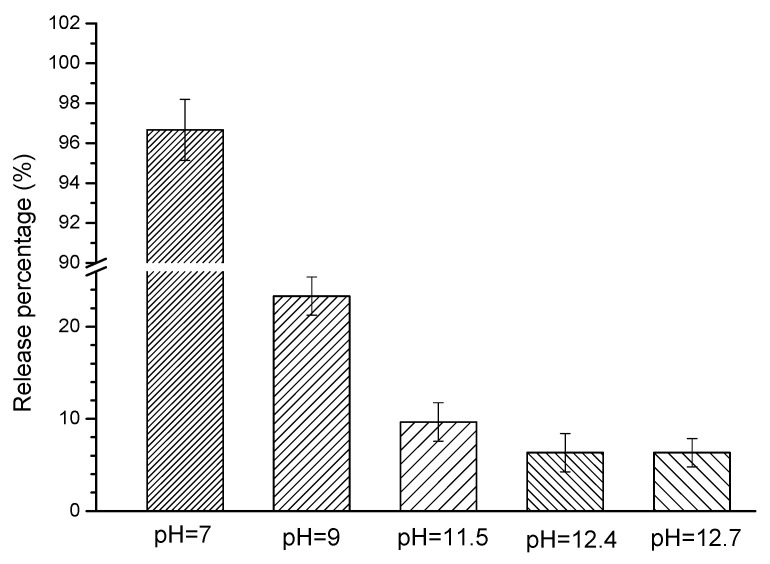
The release percentage of BTA reserved in MCI from mortar sample after immersed in SPS with different pH for 3 days.

**Figure 4 materials-14-05517-f004:**
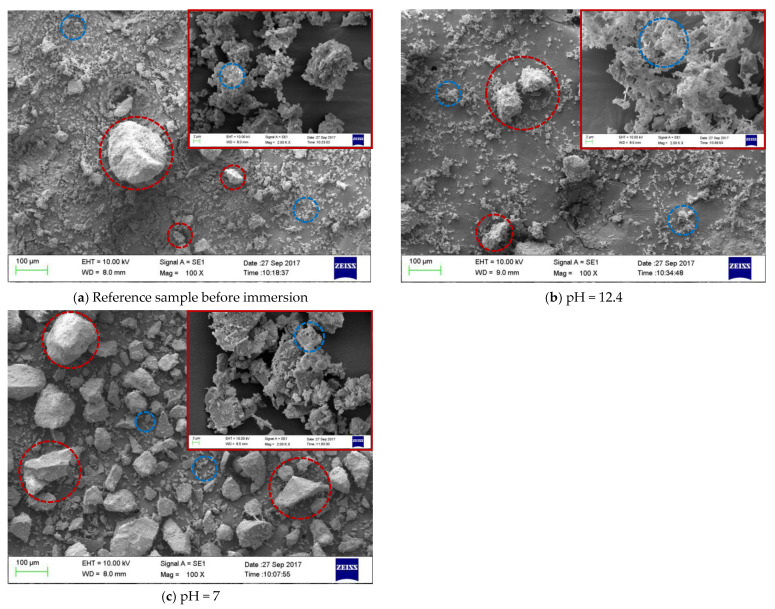
SEM images of mortar samples after immersed in different SPS for 3 days: (**a**) Reference sample before immersion; (**b**) pH = 12.4; (**c**) pH = 7.

**Figure 5 materials-14-05517-f005:**
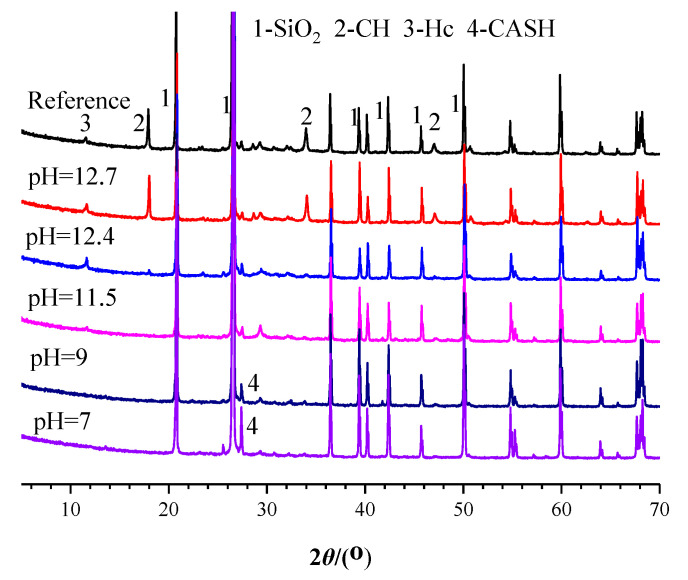
XRD patterns of the mortar samples after immersed in SPS with different pH for 3 days.

**Figure 6 materials-14-05517-f006:**
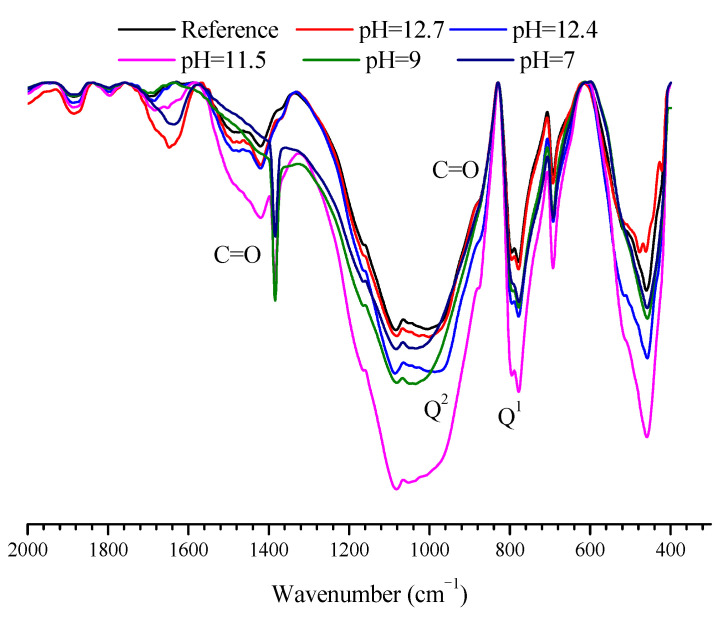
FTIR curves of the mortar samples after immersed in SPS with different pH for 3 days.

**Figure 7 materials-14-05517-f007:**
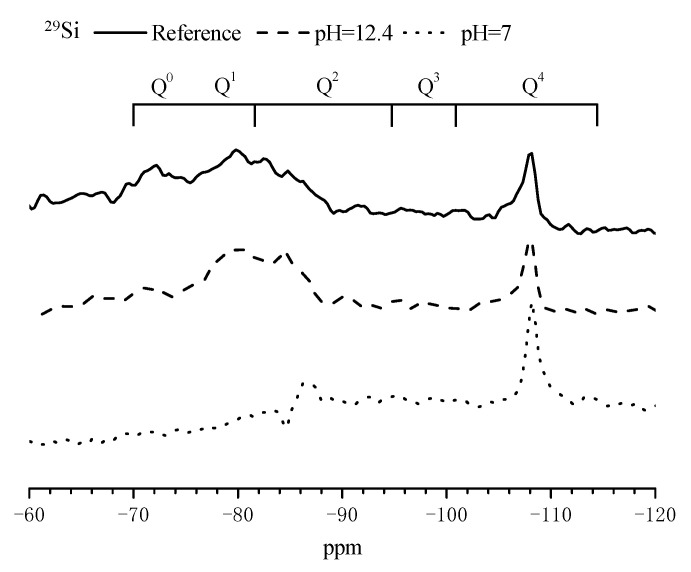
^29^Si NMR spectra of the mortar samples after immersed in different SPS for 3 days.

**Figure 8 materials-14-05517-f008:**
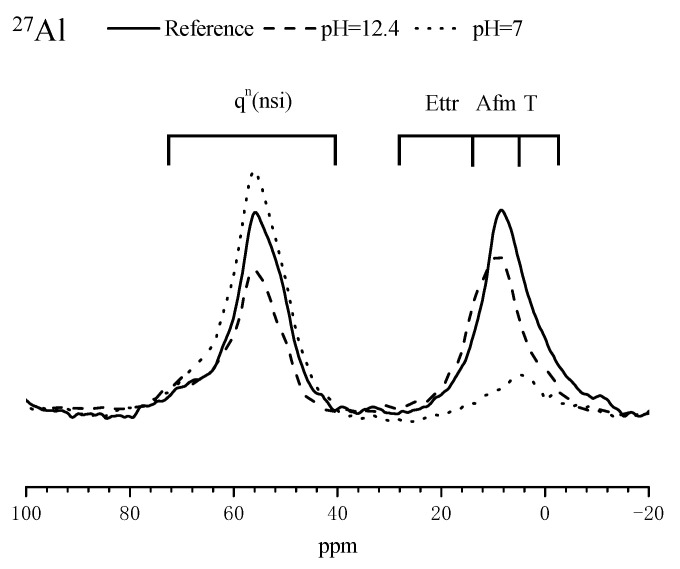
^27^Al NMR spectra of the mortar samples after immersed in different SPS for 3 days.

**Figure 9 materials-14-05517-f009:**
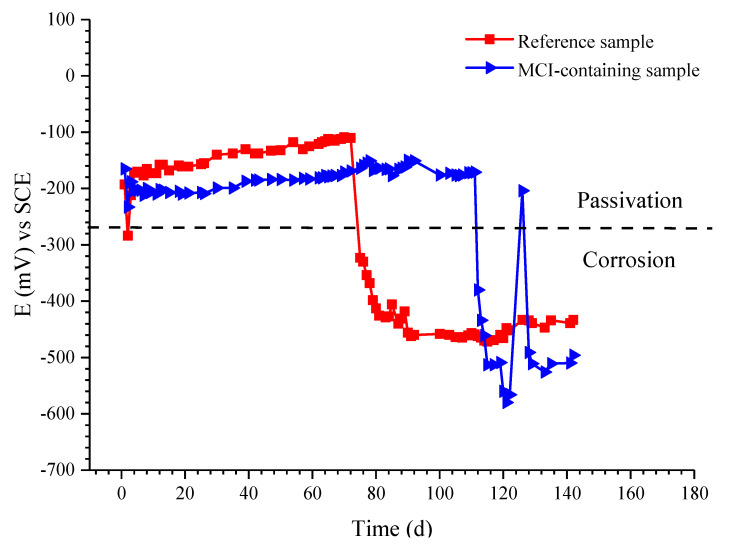
Open circuit potential of the reinforced mortar immersed in 3.5 wt. % NaCl solution.

**Figure 10 materials-14-05517-f010:**
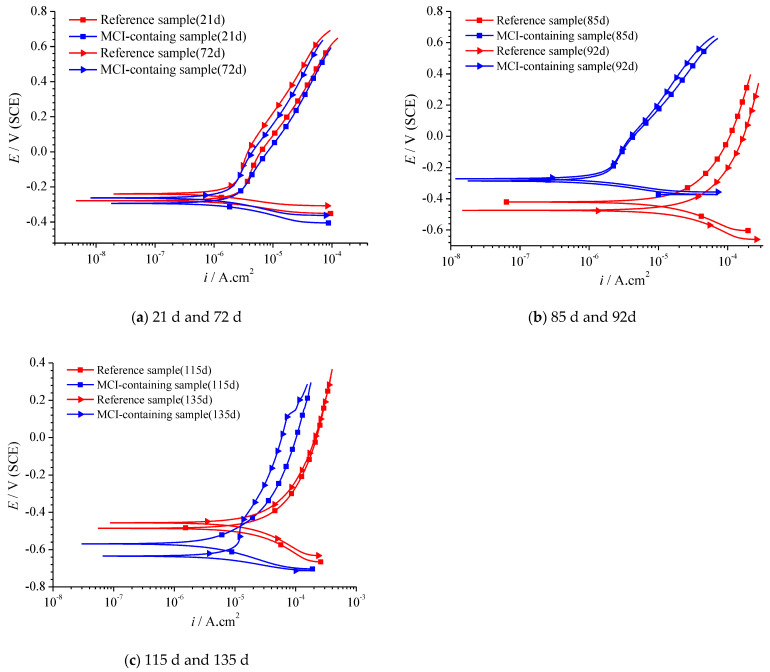
Potentio-dynamic polarization curves of reinforced mortar in 3.5% NaCl solution at different immersion ages: (**a**) 21 d and 72 d; (**b**) 85 d and 92d; (**c**) 115 d and 135 d.

**Figure 11 materials-14-05517-f011:**
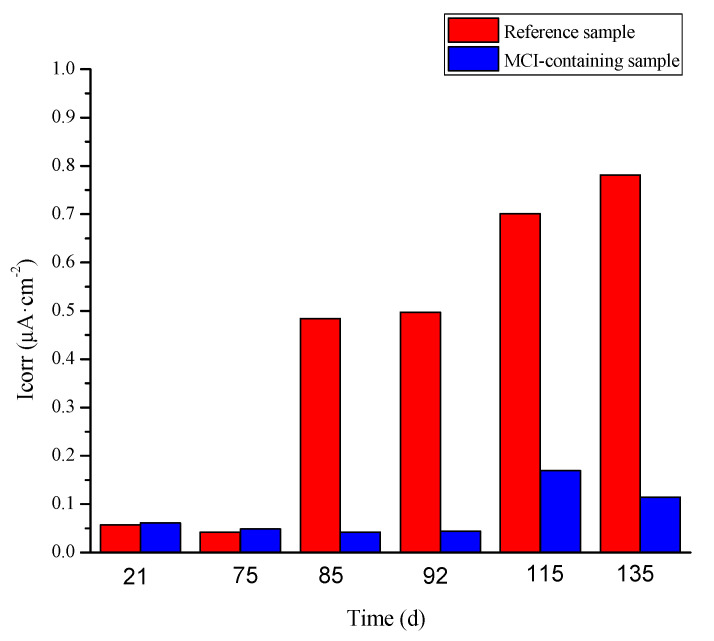
Corrosion current density of the embedded reinforcement calculated based on PD curves.

**Figure 12 materials-14-05517-f012:**
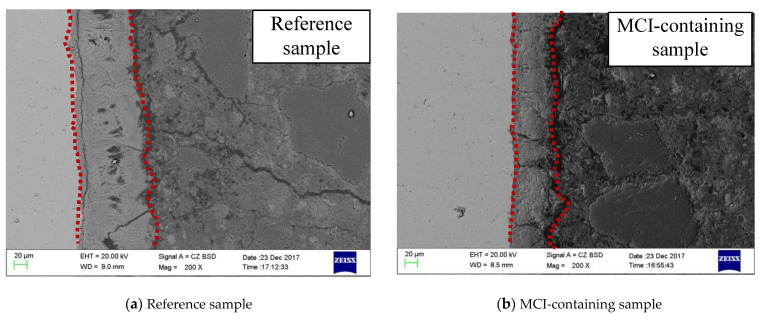
SEM images at the reinforcement/mortar interface after the sample was immersed in 3.5% NaCl solution for 135 days: (**a**) Reference sample; (**b**) MCI-containing sample.

**Figure 13 materials-14-05517-f013:**
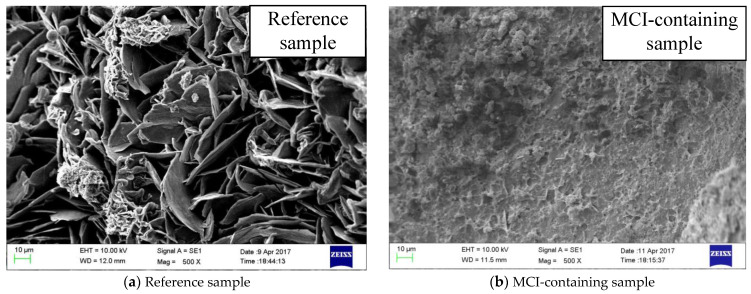
SEM images of the reinforcement surface for different samples after immersed in NaCl solution for 135 days: (**a**) Reference sample; (**b**) MCI-containing sample.

**Figure 14 materials-14-05517-f014:**
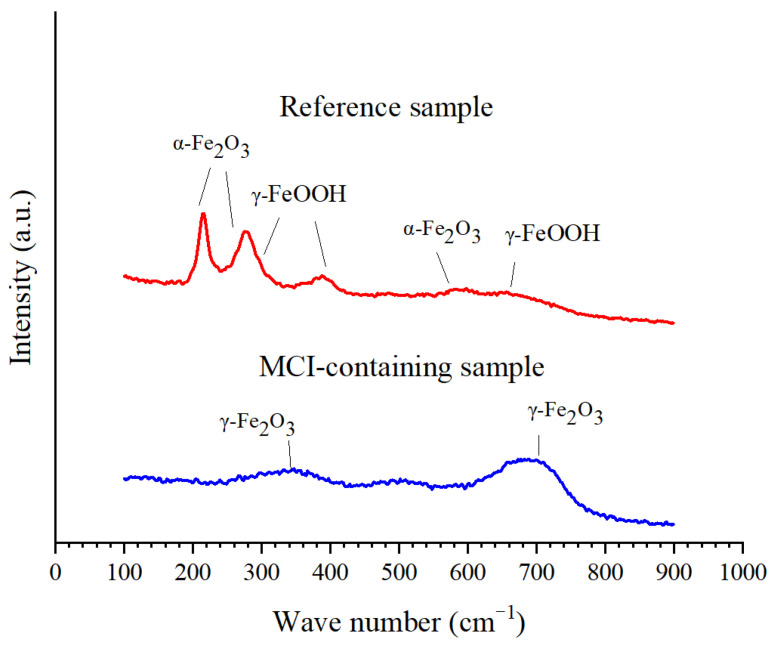
Raman spectrum of the formed corrosion products for different samples after immersed in 3.5% NaCl solution for 135 days.

**Figure 15 materials-14-05517-f015:**
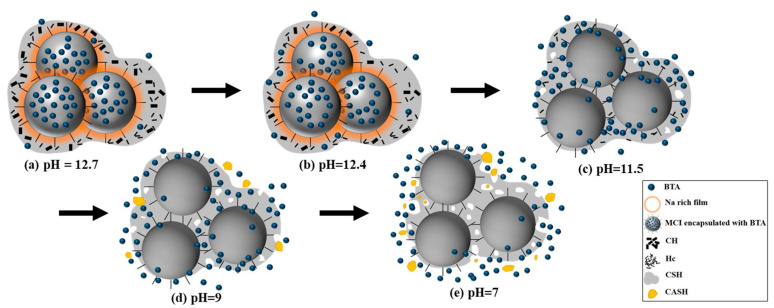
Release process of the reserved BTA in mortar sample under different pH environment: (**a**) pH = 12.7; (**b**) pH = 12.4; (**c**) pH = 11.5; (**d**) pH = 9; (**e**) pH = 7.

**Table 1 materials-14-05517-t001:** Chemical composition of PII 42.5 Portland cement (wt. %).

SiO_2_	Al_2_O_3_	Fe_2_O_3_	CaO	MgO	K_2_O	Na_2_O	SO_3_	Others	LOI *
21.60	4.35	2.95	63.81	1.76	0.51	0.16	2.06	1.61	1.19

* LOI, loss on ignition.

**Table 2 materials-14-05517-t002:** Mineral composition of P·II 42.5 Portland cement (wt. %).

C_3_S *	C_2_S *	C_3_A *	C_4_AF *	CaSO_4_·2H_2_O	CaCO_3_
56.20	19.61	6.54	8.97	4.33	4.35

* Calculated by Bogue method [21].

**Table 3 materials-14-05517-t003:** Chemical composition of HPB235 construction steel used in this study.

Elements Composition (wt. %)					
C	Si	Mn	S	P	Fe
0.18	0.28	0.55	0.04	0.04	98.91

**Table 4 materials-14-05517-t004:** Chemical composition (wt. %) of the mortar sample after immersed in SPS with different pH for 3 days.

Oxides	Reference	pH = 12.7	pH = 12.4	pH = 11.5	pH = 9.0	pH = 7.0
SiO_2_	82.1	82.7	85.9	89.0	89.1	92.0
CaO	12.3	12.2	8.8	6.0	5.9	2.9
Al_2_O_3_	2.9	2.7	2.9	3.0	3.0	2.9
Fe_2_O_3_	1.1	1.0	1.1	1.0	1.0	1.0
SO_3_	0.6	0.7	0.5	0.3	0.4	0.4
K_2_O	0.6	0.6	0.5	0.5	0.6	0.6
TiO_2_	0.1	0.1	0.1	0.1	0.0	0.0
Others	0.3	0.0	0.2	0.1	0.0	0.2

**Table 5 materials-14-05517-t005:** Position of ^29^Si NMR sites in hardened Portland cement paste [29].

Sites	Chemical Shift (ppm)
C_2_S-Q^0^	−71
C_3_S-Q^0^	−69 to −73
CASH-Q^1^	−79
CASH-Q^2^(1Al)	−82
CASH-Q^2^	−85
Q^2^(1Al)	−86
Q^2^	−92
Q^3^(1Al)	−95
Q^3^	−101
Q^4^(1Al)	−104
Q^4^	−110

## Data Availability

Data available on request due to restrictions e.g., privacy or ethical.

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
