# Peer review of "pH-Triggered Release Performance of Microcapsule-Based Inhibitor and Its Inhibition Effect on the Reinforcement Embedded in Mortar"

_materials, 2021, doi:10.3390/ma14195517_

Round 1

Reviewer 1 Report

This manuscript reports the experimental results of the release behavior of benzotriazole (BTA) in microcapsule-based inhibitors in mortar samples, further clarifying the influence of different hydration products on the release process and evaluating the corrosion inhibition effect of MCI on the reinforcement embedded in mortar. The results are interesting. The text seems to be quite clear and well written. There are some minor comments as mentioned below which should be addressed.

What are the effects of MCI solution on the mechanical properties of the studied mortar samples?

Line 112-113, does ethanol affect the release of BTA?

Line 256, the infrared peak corresponding to Q1 with pH of 9 and 7 did not disappear.

Line 306, it is not “slightly”.

Fig. 8, please give a more detailed explanation for the MCI-containing sample.

Author Response

Please see the detailed responses to the comments in the attached file.

Reviewer 2 Report

Thank you for this contribution. This is an interesting and timely manuscript that discusses the adverse impact of corrosion mortar reinforcement. The conducted analysis is typically standard and falls within the expected work from such a publication and hence the work merits publication. As such, the authors are invited to properly address the following minor items:

  1. In general, the introduction is light and does not represent the state of the art in this domain. The amount of works in this area continues to rapidly rise. The authors are advised to strengthen their literature review section with supplementary material. 

  2. ".. the amount of hydration products was further reduced and mainly sands with large dimensions were observed in SEM image" Please articulate this phenomenon

Author Response

(The authors gave the same response as above.)

Reviewer 3 Report

 Authors studied very specific topics:  the release of benzotriazole  in microcapsule-based inhibitors  in mortar sample 
to clarify the influence of different hydration products on the release process.   Microcapsules carrying with self-healing agents were proposed for  achieving smart corrosion protection of reinforced concrete.

The work solves a very specific and important matter but lacks the scientific generalization that the journal of quality and scope of the Material definitely requires. The whole work must provide the work with a greater overlap for the readers of the material magazine. The detailed results should therefore be placed in the broader context of the material changes under specific conditions  research.

Author Response

(The authors gave the same response as above.)

Round 2

Reviewer 3 Report

To be published.